# MCUCoder: Adaptive Bitrate Learned Video Compression for IoT Devices

## Abstract

The rapid growth of camera-based Internet of Things (IoT) devices demands the need for efficient video compression, particularly for edge applications where devices face hardware constraints, often with only 1 or 2 MB of RAM and unstable internet connections. Traditional and deep video compression methods are designed for high-end hardware, exceeding the capabilities of these constrained devices. Consequently, video compression in these scenarios is often limited to Motion-JPEG (M-JPEG) due to its high hardware efficiency and low complexity. This paper introduces MCUCoder, an open-source adaptive bitrate video compression model tailored for resource-limited IoT settings. MCUCoder features an ultra-lightweight encoder with only 10.5K parameters and a minimal 350KB memory footprint, making it well-suited for edge devices and Microcontrollers (MCUs). While MCUCoder uses a similar amount of energy as M-JPEG, it reduces bitrate by 55.65% on the MCL-JCV dataset and 55.59% on the UVG dataset, measured in MS-SSIM. Moreover, MCUCoder supports adaptive bitrate streaming by generating a latent representation that is sorted by importance, allowing transmission based on available bandwidth. This ensures smooth real-time video transmission even under fluctuating network conditions on low-resource devices. Source code available at [Link removed due to double-blind policy, code submitted in ZIP].

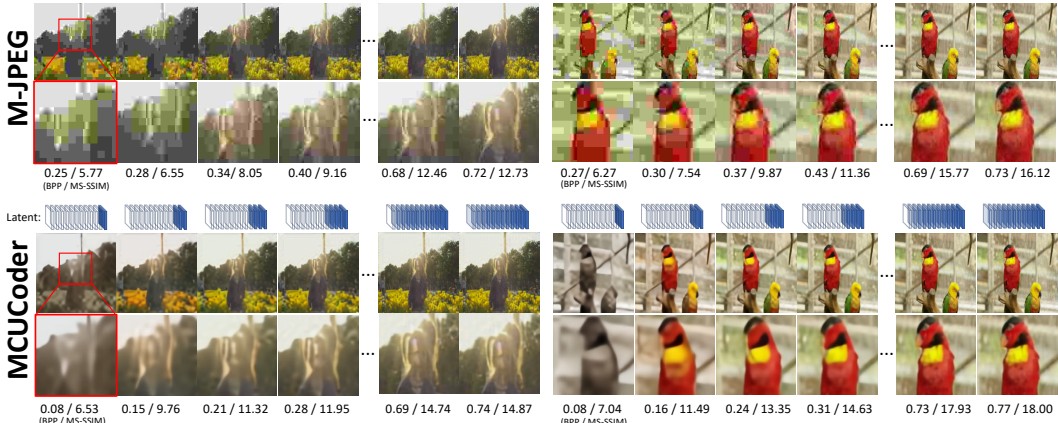

Figure 1: Qualitative comparison of MCUCoder and M-JPEG across various compression rates on two videos from the MCL-JCV (Wang et al., 2016) and UVG (Mercat et al., 2020) datasets. As we can see, MCUCoder offers a significantly better MS-SSIM/bpp trade-off. For instance, at 0.15 bpp in the left example, with MCUCoder we can see the person's face whereas with M-JPEG we need at least 0.34 bpp to make out the face. Note that the images in each column do not necessarily have the same bitrate. More examples are reported in Appendix A.

## 1 Introduction

**Motivation:** The number of camera-based IoTs devices using always-on MCU is growing rapidly, reaching tens of billions (Lin et al., 2020). These devices are widely used in applications such as

surveillance cameras (Hu et al., 2020; Josephson et al., 2019; Naderiparizi et al., 2018), wearable cameras (Veluri et al., 2023), robotics (Nakanoya et al., 2023), wildlife monitoring (Iyer et al., 2020), road monitoring (Hojjat et al., 2024), and smart farming (Koh et al., 2021). Typically, they capture raw frames through a camera sensor, encode them, and transmit the compressed version to a server via the Internet for further processing, including human observation or AI tasks such as object detection and classification (Yao et al., 2020). Therefore, a video encoder is necessary to efficiently compress the captured frames before transmission. However, in IoT environments, there are two primary limitations: constrained hardware resources and limited communication bandwidth.

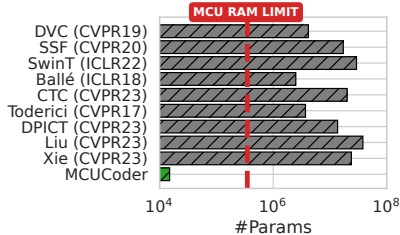

Figure 2: Number of parameters of `MCUCoder` and other learned image compression (Ballé et al., 2018; Toderici et al., 2017; Lee et al., 2022; Jeon et al., 2023; Liu et al., 2023; Xie et al., 2021; Zhu et al., 2022) and video compression models (Agustsson et al., 2020; Lu et al., 2019).

**1 - Limited Hardware:** Although traditional video codecs like H.264 (Wiegand et al., 2003), H.265 (Sullivan et al., 2012), and the newer H.266 (Bross et al., 2021) provide excellent performance, they demand significant hardware for extracting the intra and inter-frame correlations. For example, H.265 encoding involves highly computationally intensive tasks such as motion estimation with sub-pixel accuracy, Rate Distortion Optimization (RDO) for choosing optimal intra-prediction modes, and Context Adaptive Binary Arithmetic Coding (CABAC) for entropy coding. Additionally, a single video frame at $224 \times 224$ resolution requires about 150 KB of RAM, which is a lot for the low-cost, low-energy MCUs used in IoT devices that typically have only 1–2 MB of RAM. Consequently, inter-frame compression or any other kind of multi-frame analysis is not practically feasible on such constrained devices. Similarly, while Neural Networks (NNs) and AI-based compression methods outperform traditional models (Agustsson et al., 2020; Lu et al., 2019), they also often require considerable RAM and GPU resources. For instance, just storing a model with 1M parameters requires around 4 MB of RAM; see Fig. 2. As a result, in such settings, devices are typically limited to using M-JPEG (Pennebaker and Mitchell, 1992), a video compression format where each frame is compressed individually as a JPEG image, which is efficient and hardware-friendly.

**2 - Limited Internet:** Many IoT devices are located in remote areas where Internet connection is weak and unstable, making it necessary for the encoder to have an **Adaptive Bitrate Encoding** that can generate video streams with varying bitrate. This feature allows the encoder to dynamically adjust its quality according to the available bandwidth, ensuring continuous and smooth playback. This is especially important for real-time applications like live monitoring, where it is crucial to avoid interruptions and maintain a consistent user experience despite fluctuating network conditions. However, implementing an adaptive bitrate encoder adds complexity, as it requires mechanisms to prioritize bit stream information based on its impact on frame quality (e.g., PSNR or MS-SSIM), which is challenging for constrained devices.

**Approach:** To address these challenges, we introduce `MCUCoder`, *an adaptive bitrate deep video compression model tailored for resource-limited IoT devices*. Our approach focuses on creating an "asymmetric" compression model that features an ultra-lightweight encoder designed to be both computationally efficient and memory-friendly. Also, `MCUCoder` produces an "adaptive bitrate" bitstream. Specifically, in `MCUCoder`, we train the encoder using stochastic dropout such that, instead of explicitly detecting the important parts, it produces latent channels that are sorted based on importance. Afterward, based on the available internet bandwidth, the encoder transmits the first $k$ channels to the decoder; see Fig. 1. This approach is beneficial for low-power MCUs since it shifts the complexity of identifying important data to the training phase rather than the inference phase. Also, by employing stochastic dropout training, the decoder can reconstruct the frame even with partial data availability, which is essential for maintaining smooth and uninterrupted video transmission in real-time applications, where network conditions can vary. Additionally, `MCUCoder`'s encoder is INT8 quantized, allowing it to utilize Digital Signal Processor (DSP) and CMSIS-NN (ARM-software, 2024) accelerators for faster processing and reduced power consumption.

Figure 3: Overview of `MCUCoder` architecture. With stochastic dropout training, the encoder compresses the input frame into a sorted latent space. Subsequently, channels are independently quantized and transmitted according to the available bandwidth. The decoder reconstructs the frame by zeroing out missing channels.

**Contributions:**

1. `MCUCoder` has an ultra-lightweight encoder with only 10.5K parameters and a minimal memory footprint of roughly 350KB RAM on nRF5340 and STM32F7 MCUs, making it suitable for such low-resource IoT devices.

2. `MCUCoder` has an energy-efficient INT8 quantized encoder, which leverages the MCU's DSP and CMSIS-NN accelerators to achieve JPEG-level energy efficiency. Compared to its main baseline, M-JPEG, it saves 55.65% overall bit rate on the MCL-JCV dataset and 55.59% on the UVG dataset, measured in MS-SSIM.

3. `MCUCoder` produces a progressive bitstream that enables adaptive bitrate streaming, allowing robust video transmission under varying network conditions.

## 2    RELATED WORK

In this section, we provide an overview of both traditional and NN based video compression techniques, as well as video compression methods tailored specifically for IoT environments.

### 2.1    TRADITIONAL AND NN BASED VIDEO COMPRESSION

Video compression is a field that has been evolving for decades. Beyond traditional codecs like H.264 (Wiegand et al., 2003), H.265 (Sullivan et al., 2012), and H.266 (Bross et al., 2021), deep learning-based approaches often replace conventional modules such as motion compensation (Agustsson et al., 2020; Yang et al., 2020), transform coding (Zhu et al., 2022; Gao et al., 2021), and entropy coding (Xiang et al., 2023; Mentzer et al., 2022). Also, some work has been done regarding the end-to-end optimization of video compression models (He et al., 2020; Van Rozendaal et al., 2021; Khani et al., 2021).Lu et al. (2019) introduce DVC, the first end-to-end deep video compression model. Hu et al. (2022; 2021) extend DVC to operate in both pixel and feature domains. Li et al. (2021) and Liu et al. (2020) reduce bitrates by modeling probabilities over video frames using conditional coding. Also, in recent years, there has been growing interested in using implicit neural representations for video compression (Kwan et al., 2024; Chen et al., 2021). However, due to their substantial hardware requirements, these models are unsuitable for deployment on low-resource IoT devices.

### 2.2    VIDEO COMPRESSION FOR IOT

We can categorize IoT-based video encoders into two parts: hardware-based and software-based. Hardware approaches primarily focus on designing more power-efficient camera sensors (Morishita et al., 2021; Ji et al., 2016; Bejarano-Carbo et al., 2022) and more efficient MCU circuits and processors (Lefebvre et al., 2021; Rossi et al., 2021; Xu et al., 2020). Due to its simplicity, scalability, low latency, and very low energy consumption, the most common software-based video encoder on IoT devices is M-JPEG (Pennebaker and Mitchell, 1992). Nevertheless, there have been few works exploring alternative software-based models: Veluri et al. (2023) employ M-JPEG on the encoder to capture black-and-white and colorized frames at two different resolutions and uses super-resolution

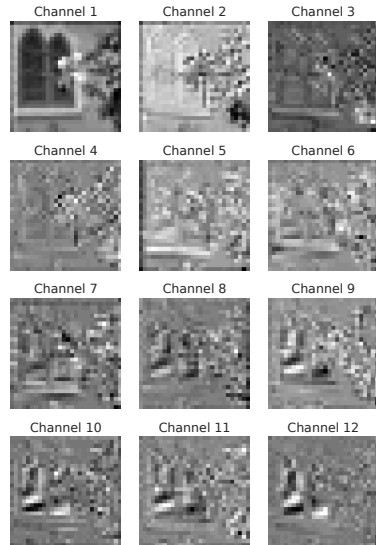

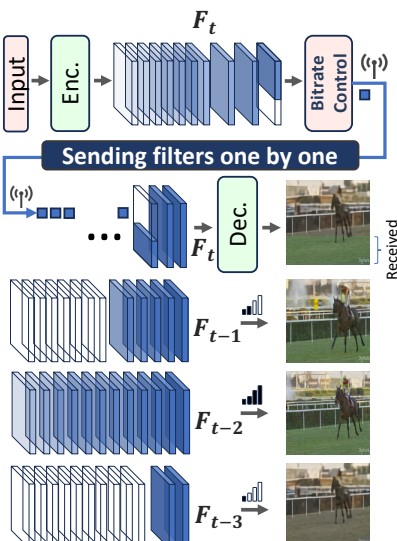

Figure 4: `MCUCoder` latent channels: Early channels (important ones) capture low-frequency features, while later channels capture high-frequency features, similar to the DCT in JPEG.

Figure 5: An example of `MCUCoder` bitrate adaptation under dynamic network bandwidth, where the bitrate control module acts as a gate to determine the number of channels to send.

methods to interpolate and colorize frames on the decoder. However, unlike `MCUCoder`, it is not adaptive and relies on a JPEG encoder on MCUs. Hu et al. (2020) propose a deep image encoder model for MCUs, but it is also non-adaptive. Additionally, they patchify the input, which significantly increases encoding time, making it impractical for real-time video compression. `MCUCoder` combines the advantages of both worlds: it offers the adaptive bitrate feature of more complex encoders, while maintaining the efficiency necessary for low-resource devices, making it an ideal solution for IoT video compression.

## 3 MCUCODER

In this section, we introduce `MCUCoder`, an adaptive bitrate asymmetric video compression model, specifically designed for IoT settings. We begin by detailing the asymmetric encoder-decoder architecture of `MCUCoder`, including the customized quantization processes. Then, we present the stochastic dropout training method, which trains the encoder of `MCUCoder` to store information in its channels based on importance.

### 3.1 ASYMMETRIC COMPRESSION

MCUs are characterized by highly constrained hardware resources, such as limited RAM, CPU, FLASH, and power availability. Additionally, existing MCU-specific NN frameworks like TFLite Micro support only a limited set of NN layers (Hu et al., 2020). To address these constraints, we propose an asymmetric (Yao et al., 2020) encoder-decoder architecture optimized for constrained devices. Due to hardware constraints, `MCUCoder` encodes each frame independently, as inter-frame compression is not feasible. The encoder contains only 10.5K parameters, while the decoder utilizes approximately 3M parameters and leverages SOTA image decompression blocks; see Fig. 3. The encoding process begins by passing input frame $f_t$ through three convolutional layers. To maximize the data range for subsequent quantization, no activation function is applied in the final encoder layer, avoiding the negative truncation caused by ReLU. Afterward, each channel of the latent is quantized into INT8 individually, followed by a further reduction to 5-bit precision to enhance compression efficiency. For the decoder, inspired by He et al. (2022), we integrate a combination of attention blocks (Cheng et al., 2020) and residual bottleneck blocks (He et al., 2016) to reconstruct the frame; see Fig. 3.

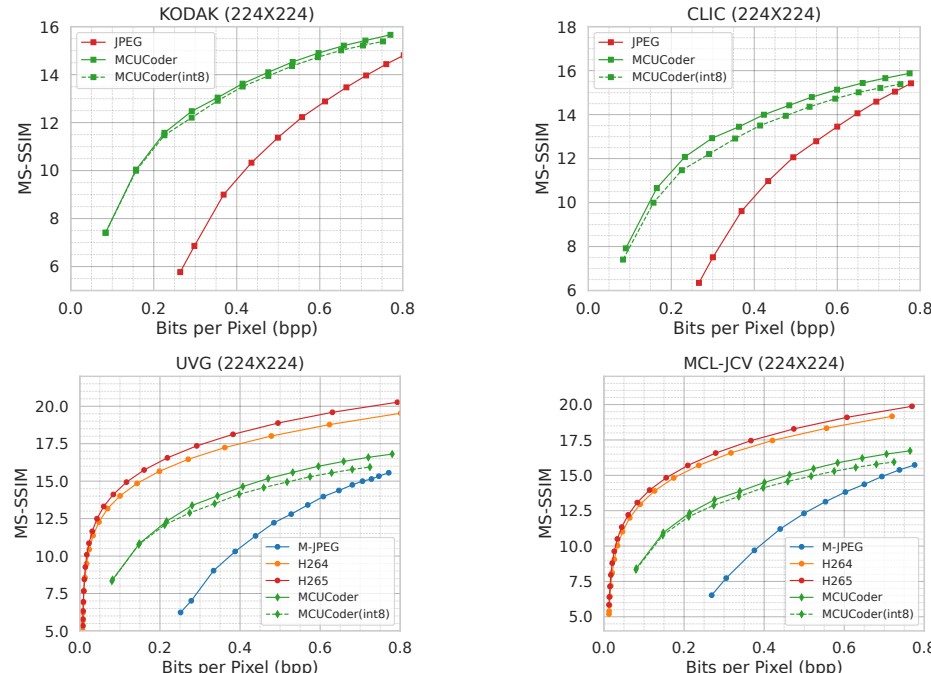

Figure 7: Comparison of `MCUCoder` (quantized and non-quantized model) and baselines on the image (KODAK (Eastman Kodak, 1993), CLIC (cli, 2020)) and video (MCL-JCV (Wang et al., 2016), UVG (Mercat et al., 2020)) compression datasets. For context, we also compare with H.264 and H.265 on video datasets, despite being impractical for MCUs due to high hardware demands. All datasets are resized to $224 \times 224$.

## 3.2 STOCHASTIC DROPOUT TRAINING

Bitrate adaptation is a feature that typically introduces additional complexity to the encoding process, which can be challenging to implement on MCUs due to resource constraints. In the literature, dropout (Srivastava et al., 2014) serves as a powerful tool for enhancing generalization in NNs. Building on this insight, we employ a "biased" version of dropout to train `MCUCoder` in a way that instead of random dropping, it drops from the tail of the latent (Hojjat et al., 2023). Specifically, on each iteration, after the encoder $E$ gets the input frame $f_t$, it generates the latent representation $z_N$, where $N$ is the number of the channels of the latent. Afterward, from a uniform distribution, denoted as $\mathcal{U}_{(0,1)}$, it generates a number, denoted as $k$, and drops (zero out) the last $\lfloor k \times N \rfloor$ channels from $z_N$. As a result, instead of $z_N$, the decoder $D$ gets $z_{[0:\lfloor k \times N \rfloor]}$, fills the missing channels with zero, and then reconstructs the output.

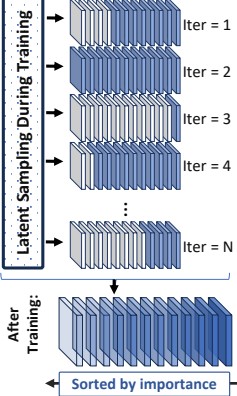

$$f_t \rightarrow E(f_t) \rightarrow z_N \xrightarrow{k \sim \mathcal{U}_{(0,1)}} z_{[0:\lfloor k \times N \rfloor]} \rightarrow D(z_{[0:\lfloor k \times N \rfloor]}) \rightarrow \hat{f}_t \quad (1)$$

Figure 6: Stochastic dropout training

This tailored version of dropout biases the training to prioritize the earlier channels over the later ones. Consequently, the encoder learns to encode more critical information (low frequency) in the initial feature maps and less important (high frequency) details in the subsequent ones; see Fig 6. This prioritization enables flexible bitrate adaptation: upon encoding each frame, the encoder starts transmitting the most significant channels first. Depending on the available bandwidth, the bitrate control module determines how many channels need to be sent to the decoder to ensure uninterrupted streaming; see Fig 5. Importantly, because the latent features are pre-ordered by significance, the bitrate control module basically acts like a simple gate and does not add any extra computational complexity to the encoder.

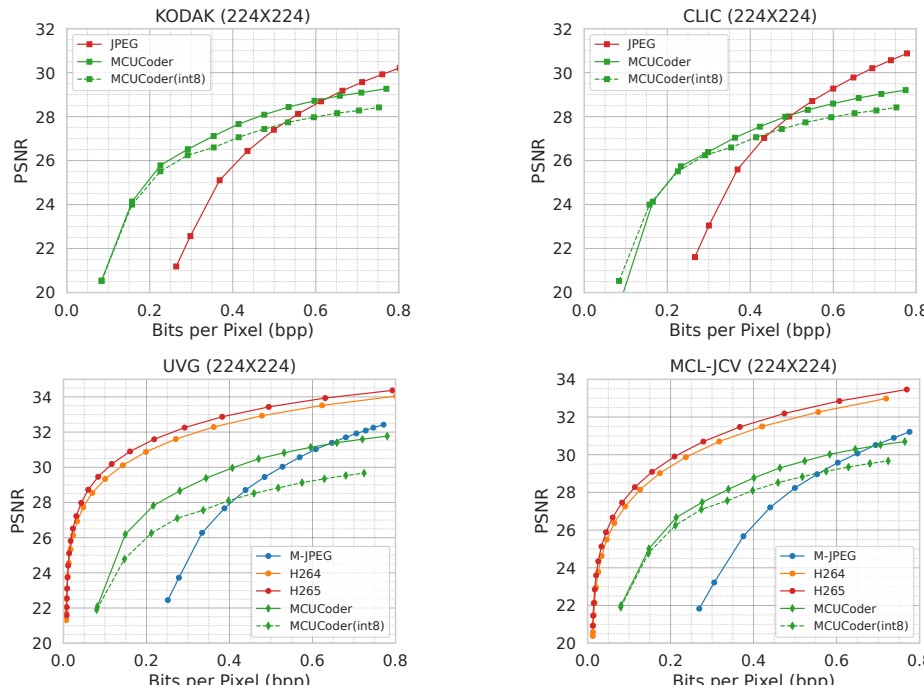

Figure 8: Comparison of `MCUCoder` (quantized and non-quantized model) and baselines on the image (KODAK (Eastman Kodak, 1993), CLIC (cli, 2020)) and video (MCL-JCV (Wang et al., 2016), UVG (Mercat et al., 2020)) compression datasets. For context, we also compare with H.264 and H.265 on video datasets, despite being impractical for MCUs due to high hardware demands. `MCUCoder` is designed for IoT environments, prioritizing structural integrity over fine detail and therefore it is optimized for MS-SSIM. In contrast, JPEG optimizes for PSNR, which is why M-JPEG performs slightly better in PSNR at higher bitrates.

## 4 EVALUATION

This section presents a comprehensive evaluation of `MCUCoder` across both image and video compression tasks. We compare its performance against JPEG,M-JPEG, and traditional codecs, with a focus on metrics such as MS-SSIM, PSNR, and BD-rate. Additionally, we analyze `MCUCoder`'s efficiency on resource-constrained MCU devices, highlighting its computational and energy performance.

### 4.1 SETTINGS

We train `MCUCoder` on the 300K largest ImageNet images (Deng et al., 2009) and apply noise-downsampling preprocessing (He et al., 2021; Ballé et al., 2018). We use Adam with an initial learning rate of $10^{-4}$ and a batch size of 16, and train for 1M iterations, lowering the learning rate to $10^{-5}$ in the final 50K iterations (He et al., 2022). To address quantization effects, we add random noise to the latent. Since `MCUCoder` is specifically designed for IoT environments, where the structure of the output is more critical than fine details, we use MS-SSIM as the loss function. We also quantize inputs, weights, and activations to INT8 for RAM efficiency and to leverage DSP and CMSIS-NN accelerators (ARM-software, 2024) in MCUs. We use post-training quantization existing in TFLite-Micro (TensorFlow, 2023) to reduce latency, processing power, and model size with minimal degradation in model accuracy. For all comparisons, we report performance metrics for both the FLOAT32 and INT8 models.

Table 1: MCUCoder (Quantized) BD-rate results. The anchor is M-JPEG.

| Dataset | MS-SSIM | PSNR |
|---|---|---|
| MCL-JCV | -55.65% | -47.39% |
| UVG | -55.59% | -35.28% |
| KODAK | -55.75% | -43.01% |
| CLIC | -49.54% | -38.02% |

Table 2: Resource demands of MCUCode on nRF5340 and STM32F7 MCUs.

| | nRF5340 | STM32F7 |
|---|---|---|
| Exec (ms) | 1,969 | 237 |
| RAM (KB) | 344 (33%) | 360 (17%) |
| Flash (KB) | 100 (10%) | 107 (5%) |

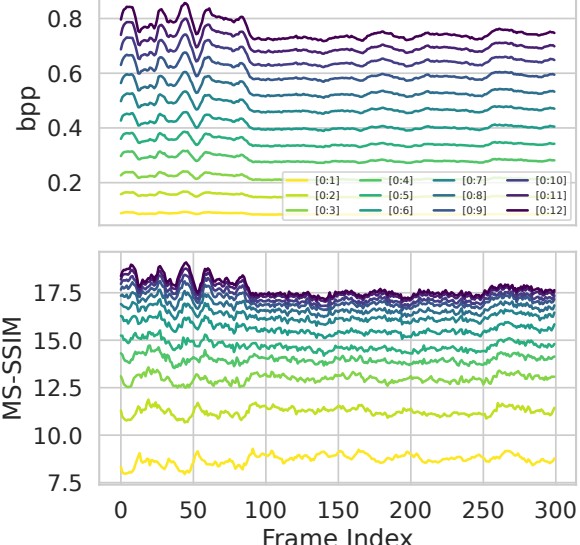

Figure 9: MS-SSIM and bpp for the SunBath video from UVG (Mercat et al., 2020) dataset. [0:k] shows the use of the first k channels (out of 12) for decoding.

## 4.2 QUANTITATIVE RESULTS

Due to the limited hardware resources of MCUs, inter-frame compression is not practically feasible. As a result, in such devices, video compression is limited to M-JPEG where each frame is compressed independently. Therefore, in addition to evaluating MCUCoder and its baselines from the perspective of video compression, we also assess its performance on image compression datasets. Given the lower resolution commonly encountered in IoT scenarios, we resize all the videos and images to $224 \times 224$.

**Video compression:** We evaluate MCUCoder on the UVG (Mercat et al., 2020) and MCL-JCV (Wang et al., 2016) datasets, comparing its performance to M-JPEG , see Fig. 7. For additional context, we include comparisons with traditional video codecs such as H.264 (Wiegand et al., 2003) and H.265 (Sullivan et al., 2012), even though these codecs are impractical for deployment on MCUs due to their significant computational and hardware demands. Also, we report the Bjøntegaard Delta (BD) rate (Bjøtegaard, 2001) for both datasets in Table 1. The results indicate that MCUCoder achieves a significantly higher MS-SSIM per bit compared to M-JPEG, highlighting its ability to deliver better video quality at lower bitrates. This is especially valuable for IoT applications, where achieving high compression rates with minimal computational overhead is crucial due to limited hardware resources. Additionally, MCUCoder has 12 "stacked" channels in its latent space, which provides 12 levels of quality that can be dynamically adjusted based on the available network bandwidth. In Fig. 9, we illustrate the bpp and MS-SSIM for each frame in a video from the UVG dataset for all 12 levels of quality. The results show that using more channels for decoding leads to a higher MS-SSIM, which verifies the effectiveness of the proposed stochastic dropout training.

**Image compression:** To assess the image compression capabilities of MCUCoder, we conduct experiments on the CLIC (cli, 2020) and KODAK (Eastman Kodak, 1993) datasets, see Fig. 7. The results in Table 1 show that MCUCoder achieves an impressive average bitrate reduction of 55.75% on the KODAK dataset and 49.54% on the CLIC dataset, compared to JPEG. As previously mentioned, MCUCoder is specifically designed for IoT environments, where preserving the structural integrity of the output is more important than capturing fine detail, leading to its optimization for MS-SSIM. In contrast, JPEG is more focused on optimizing PSNR (Wang et al., 2004), which explains why M-JPEG performs slightly better in PSNR at higher bitrates, see Fig.8.

## 4.3 LATENT ORDERING AND DCT-JPEG ALIGNMENT

Fig. 4 illustrates the 12 latent channels derived from training with the stochastic dropout method. These channels display an intriguing hierarchical structure, where the early channels capture broad,

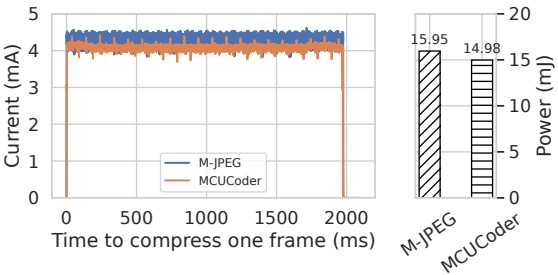

Figure 10: Energy (Millijoule) and current (Milliampere) consumption of `MCUCoder` compared to M-JPEG for compressing one frame on the nRF5340. `MCUCoder` achieves comparable energy efficiency to M-JPEG while exceeding it in BD-rate. However, the nRF5340 exhibits relatively slow processing speeds for both `MCUCoder` and M-JPEG, suggesting that its energy efficiency is better suited for event-driven applications rather than real-time streaming, where the STM32F7 excels.

low-frequency features, while the later channels progressively focus on finer, high-frequency details. This pattern closely resembles the Discrete Cosine Transform (DCT) basis matrix utilized in JPEG compression. In JPEG, the DCT plays a pivotal role in transforming image data into frequency components, allowing for efficient compression by prioritizing lower frequencies, which tend to carry more significant visual information. Similarly to `MCUCoder`, progressive JPEG leverages this frequency ordering, encoding data in a manner that allows the decoder to initially reconstruct the image using only low-frequency components, and as decoding progresses, higher-frequency details are incrementally added, resulting in a progressively refined image reconstruction.

### 4.4 STOCHASTIC DROPOUT TRAINING ANALYSIS

One potential challenge with stochastic dropout training is the risk of overfitting to specific loss functions when optimizing multiple losses concurrently. To evaluate this, we track the MS-SSIM of `MCUCoder` on the KODAK (Eastman Kodak, 1993) dataset across varying numbers of active latent channels during training. The training logs, shown in Fig. 11, demonstrate that all the sub-latents are trained in parallel without overfitting to any specific sub-latent, which verifies the effectiveness of the uniform latent sampling strategy employed in the training, see Fig. 6.

### 4.5 PERFORMANCE ON MCUS

We implement `MCUCoder` on two widely-used MCU platforms, the STM32F7 and nRF5340 MCUs, using TFLite-Micro and Zephyr RTOS. The STM32F7 features 2 MB of Flash memory, 2 MB of RAM, and a Cortex-M7 processor, while the nRF5340 is equipped with 1 MB of Flash, 512 KB of RAM, and a Cortex-M33 processor. Both MCUs support DSP and CMSIS-NN acceleration, making them well-suited for running lightweight deep learning models. As detailed in Table 2, `MCUCoder` demonstrates a low memory footprint, consuming 360 KB of RAM on the STM32F7 and 344 KB on the nRF5340, which is significantly efficient for such constrained devices. This compact memory usage highlights the suitability of `MCUCoder` for low-power, resource-constrained IoT applications. To assess the energy efficiency of `MCUCoder`, we conducted a comparative analysis against M-JPEG. Specifically, we measured the energy consumption of `MCUCoder` and an optimized JPEG encoder for the Cortex-M series[1] on the nRF5340 platform; see Fig.10. The results indicate that `MCUCoder` achieves comparable energy consumption to JPEG, while providing superior performance in terms of BD-rate, as shown in Table1. However, the nRF5340 exhibits noticeably slower processing performance compared to the STM32F7 for both `MCUCoder` and M-JPEG. This discrepancy suggests that while the nRF5340 is energy-efficient, its lower computational capabilities make it more appropriate for event-driven applications rather than real-time streaming tasks, where the STM32F7 excels.

---

[1]https://github.com/noritsuna/JPEGEncoder4Cortex-M

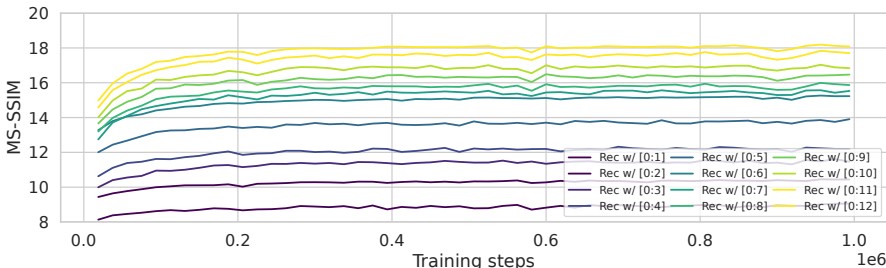

Figure 11: MS-SSIM values on the KODAK dataset during training. The notation $[0 : k]$ represents the MS-SSIM of the reconstructed image using the first $k$ latent channels out of a total of 12. As shown, with stochastic dropout training, all the sub-latents can be trained simultaneously without overfitting to any particular sub-latent.

## 5 LIMITATIONS

The design of `MCUCoder` is inherently motivated by the resource constraints typical of IoT devices; however, these constraints also constitute its limitations. One significant limitation arises from the restricted RAM available on most IoT devices, which prevents the incorporation of intra-frame compression techniques. Consequently, `MCUCoder` exhibits a performance drop when compared to more computationally demanding video compression models, such as the H.26X series, which, however, require significant hardware resources far beyond the capabilities of MCUs with only one or two MB of RAM. Furthermore, limited RAM also constrains the resolution of input frames processed by `MCUCoder`, which can negatively impact the visual fidelity of the compressed video, particularly in applications requiring higher detail. Additionally, the low clock speeds of MCU's processors, necessitated by battery conservation needs, result in prolonged encoding times for `MCUCoder`. This increased encoding duration ultimately leads to lower fps during video processing, which can hinder real-time performance and responsiveness in streaming applications. However, these limitations are not unique to `MCUCoder`. The state-of-the-art video compression model used in such constrained devices, M-JPEG, faces similar issues. M-JPEG does not utilize intra-frame compression either and requires more RAM to achieve higher resolutions, impacting visual fidelity. Like `MCUCoder`, M-JPEG's reliance on the low clock speeds of MCU processors results in longer encoding times and reduced fps. Nonetheless, despite all of these limitations, `MCUCoder` significantly outperforms M-JPEG in both image and video compression datasets.

## 6 CONCLUSION

In this paper, we presented `MCUCoder`, an open-source, ultra-lightweight video compression model designed specifically for resource-constrained IoT devices. With only 10.5K parameters and a 350KB memory footprint, compared to M-JPEG, `MCUCoder` demonstrates significant bitrate reductions—55.65% on the MCL-JCV dataset and 55.59% on the UVG dataset—while maintaining hardware efficiency similar to M-JPEG. Furthermore, `MCUCoder` supports adaptive bitrate streaming, enabling real-time video transmission under variable network conditions. These features make `MCUCoder` a promising solution for video compression in edge applications where both hardware and bandwidth are limited.

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

## A EXAMPLES OF MCUCODER

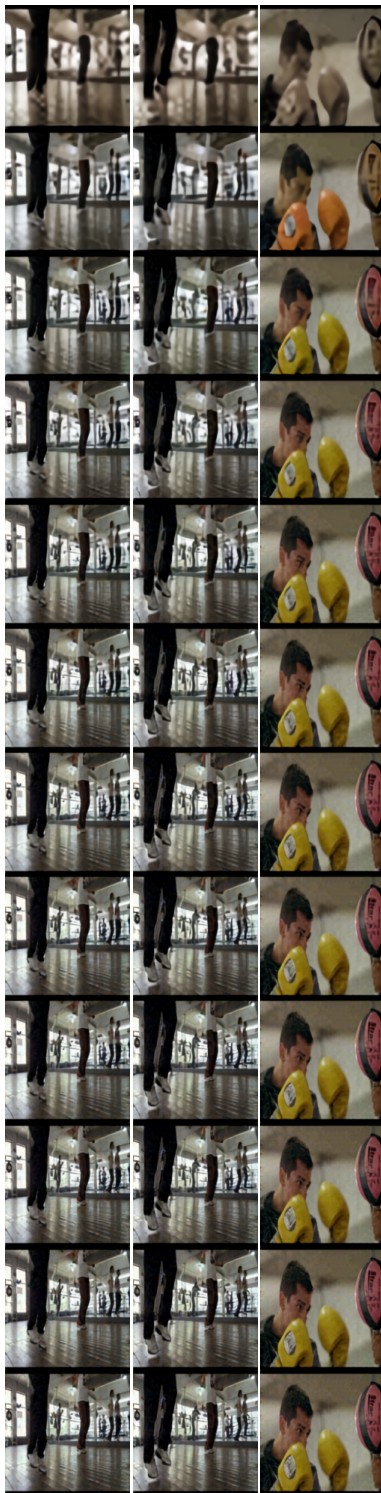 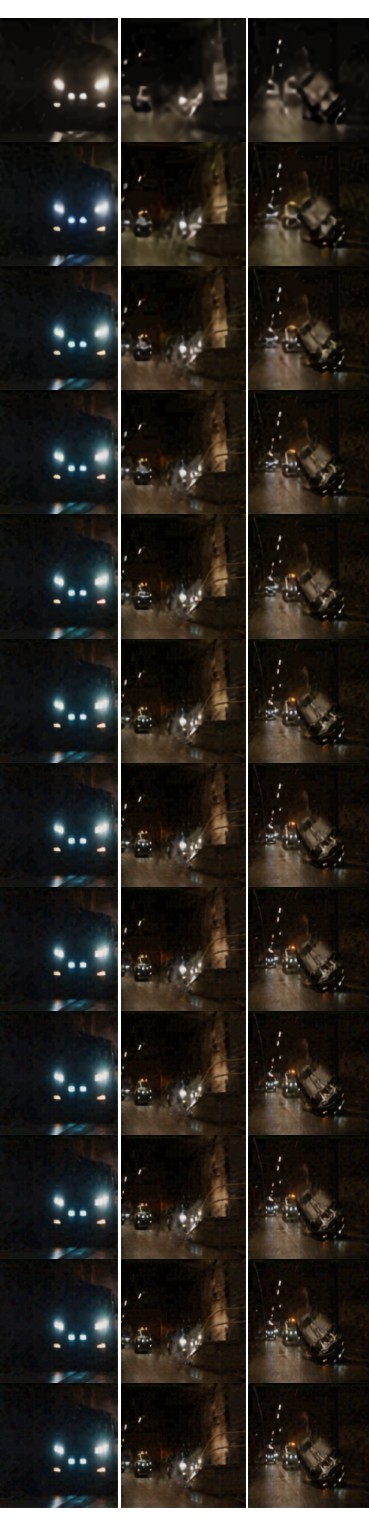

Figure 12: Some samples from the MCL-JCV Wang et al. (2016) dataset. The columns represent different frames, while the rows display progressively improving levels of quality from top to bottom, produced by MCUCoder.

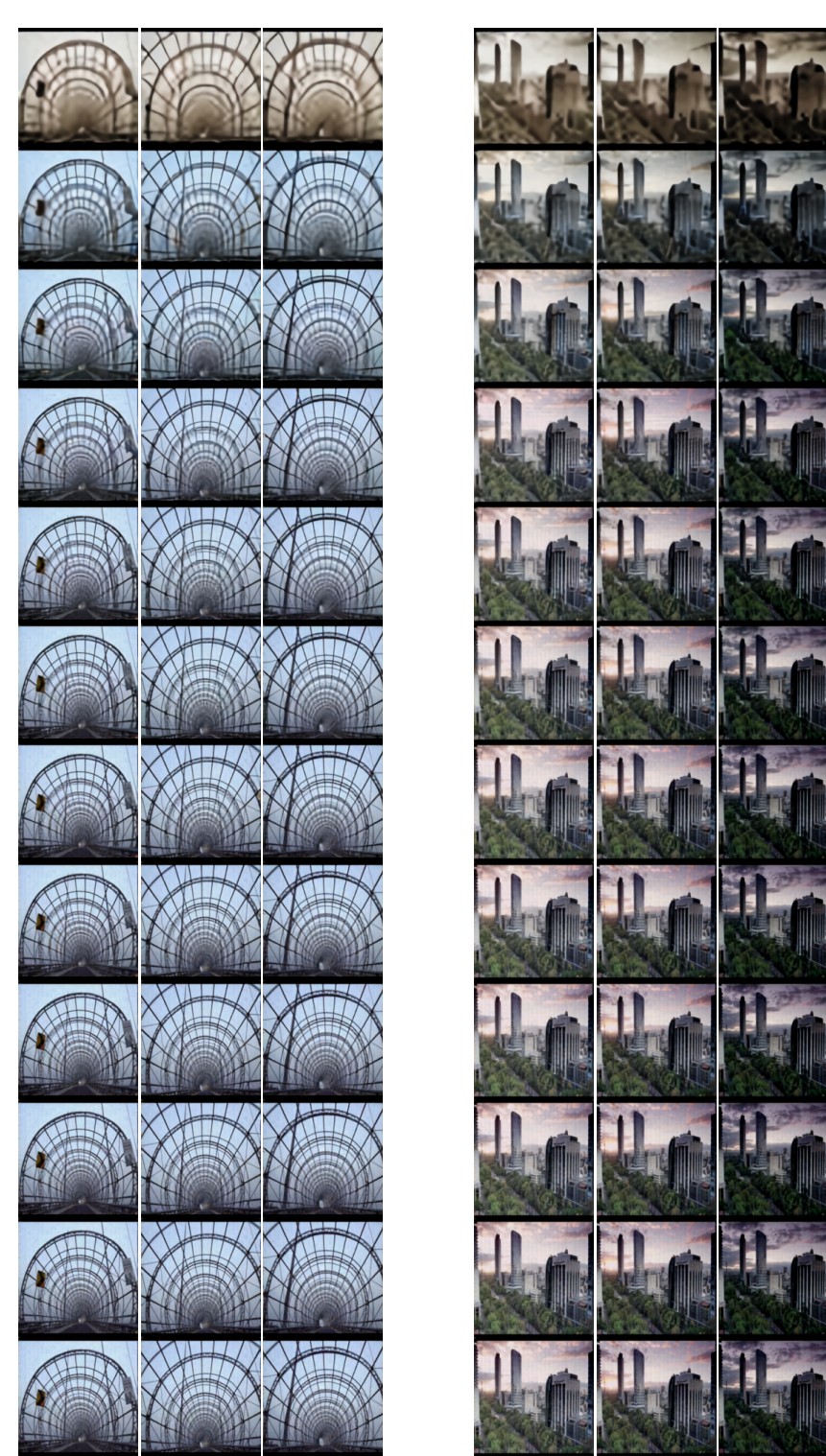

Figure 13: Some samples from the MCL-JCV Wang et al. (2016) dataset. The columns represent different frames, while the rows display progressively improving levels of quality from top to bottom, produced by `MCUCoder`.

