# OpenReview forum: "MCUCoder: Adaptive Bitrate Learned Video Compression for IoT Devices"
_ICLR.cc/2025/Conference — ICLR 2025 Conference Withdrawn Submission_

### Official Review · Reviewer_YGWh · 2024-10-21

**Soundness:** 3
**Presentation:** 2
**Contribution:** 2
**Rating:** 5
**Confidence:** 5

**Summary:**

This paper presents MCUCoder, an adaptive bitrate compression model specifically designed for resource-constrained IoT environments. MCUCoder employs an asymmetric encoder-decoder architecture to enable real-time video transmission and introduces a latent representation sorted by importance, facilitating adaptive bitrate streaming. Experimental results demonstrate that MCUCoder achieves greater bitrate savings compared to M-JPEG, while maintaining similar energy consumption.

**Strengths:**

1. The paper addresses a crucial challenge in neural compression: achieving low encoding complexity for deploying AI codecs on edge devices. It introduces an ultra-lightweight and energy-efficient INT8 quantized encoder tailored for low-resource IoT devices, which appears to be a practical solution.

2. MCUCoder leverages channel importance to generate a progressive bitstream, enabling adaptive bitrate streaming that can adjust to fluctuating network conditions.

3. The authors provide detailed information on inference time, as well as RAM and Flash memory usage, for both the encoder and decoder.

**Weaknesses:**

1. Weak compression performance: Compared to H.264, there is a significant performance gap. The authors should clarify which specific scenarios necessitate the use of extreme resource-constrained environments.

2. Missing compression baselines: In the image compression experiments, only JPEG is used as a baseline. It would be beneficial to include additional traditional compression methods such as JPEG2000 and WebP. Additionally, please clarify which traditional image compression algorithms are unsuitable for deployment on IoT devices due to resource limitations.

3. For the decoder, which is deployed in the cloud with sufficient computational resources, why not introduce an inter-frame correlation module to enhance reconstruction quality?

**Questions:**

1. When evaluating using PSNR, why didn't you train the neural codec with an MSE loss function? Using MSE could potentially lead to higher PSNR values for MCUCoder. Would this enable it to outperform M-JPEG across the entire bpp range?

2. Some methods, like Distributed DVC [1], have explored neural video compression with low encoding complexity. This method also uses an asymmetric encoder-decoder pair to accelerate encoding. I believe it should be included in the discussion.

[1] Low-complexity Deep Video Compression with A Distributed Coding Architecture.

**Details Of Ethics Concerns:**

There are no ethics concerns.

---

### Official Review · Reviewer_SnxU · 2024-11-03

**Soundness:** 3
**Presentation:** 3
**Contribution:** 3
**Rating:** 5
**Confidence:** 4

**Summary:**

This paper introduces MCUCoder, an adaptive bitrate video compression model for resource-limited IoT devices. With only 10.5K parameters and a 350KB memory footprint, MCUCoder reduces bitrate by about 55% compared to M-JPEG, while supporting smooth real-time streaming under varying network conditions.

**Strengths:**

(1) This paper explores learned video compression research on IoT Devices. For a long time, LIC could not be deployed in practical applications due to the huge consumption of resources, and the study solved the problems to some extent. I think the entry point is novel.

(2) The experiment proves that MCUCoder has obvious performance improvement compared with M-JPEG on multiple datasets.

**Weaknesses:**

(1) From the architecture in Figure 3, I observe that the reason for MCUCoder's lightweight is mainly the use of quantization methods and a simple neural network layer. I wonder what other means the author used to achieve the goal of lightweight? Because some previous works [1,2] have explored the use of quantization in learning-based compression methods, I believe that mere quantization and simple network structure design may limit the degree of innovation in this paper.

(2) In the experiments of this paper, we found that although MCUCoder's RD performance is better than JPEG, it is significantly weaker than H.264 and H.265. I am concerned that the performance bottleneck may limit the use of MCUCoder. In fact, it's my main concern.

(3) The bitrate control module appears to be an innovative aspect of this paper, but its description is not sufficiently detailed. It is unclear whether the channels are transmitted in order of their importance. In addition, can you give more details about the implementation process of bitrate control module? How does the bitrate control module control the number of channels transmitted based on bandwidth?

[1] Guo, Zongyu, et al. "Soft then hard: Rethinking the quantization in neural image compression." International Conference on Machine Learning. PMLR, 2021.

[2] Duan, Zhihao, et al. "Qarv: Quantization-aware resnet vae for lossy image compression." IEEE Transactions on Pattern Analysis and Machine Intelligence (2023).

**Questions:**

Please see Weaknesses.

---

### Official Review · Reviewer_3RTV · 2024-11-04

**Soundness:** 1
**Presentation:** 1
**Contribution:** 2
**Rating:** 3
**Confidence:** 4

**Summary:**

The paper introduces MCUCoder, an adaptive bitrate video compression model specifically designed for IoT devices with severe hardware constraints (limited RAM and unstable internet connections). It enables efficient video compression and adaptive bitrate streaming on edge devices. Experimental results show that MCUCoder achieves a 55.65% bitrate reduction over M-JPEG on the MCL-JCV dataset and 55.59% on the UVG dataset in terms of MS-SSIM, while maintaining comparable energy efficiency.

**Strengths:**

1. MCUCoder is highly optimized for low-resource IoT environments, with an encoder that requires only 350KB of RAM and achieves JPEG-level energy efficiency, making it feasible for MCU devices.
2. The model supports adaptive bitrate by sorting latent representations based on importance, enabling smooth transmission even under fluctuating network conditions.
3. The INT8 quantized encoder leverages DSP and CMSIS-NN accelerators, reducing power consumption.

**Weaknesses:**

1. The paper’s innovations are limited, as many of the techniques used are adaptations of existing methods. The novelty of the proposed model is relatively low, which could limit its contribution. The contribution part is bad presentation and organization.
2. The motivation and rationale for using stochastic dropout in training are not well-explained. Given that it is meant to achieve similar effects to DCT, it’s unclear why a more established and potentially faster method like DCT was not employed instead.
3. While MCUCoder is compared with M-JPEG and traditional codecs, there are no comparisons with other recent lightweight IoT-specific video compression methods.
4. Due to the constrained resources of MCUs, MCUCoder only processes lower resolution (224x224) frames, which limit its application in scenarios that require higher detail or clarity.
5. The paper does not provide sufficient ablation experiments to validate the effectiveness of key components, such as the asymmetric architecture, stochastic dropout, and the choice of loss functions.
6. The paper relies only on MS-SSIM and PSNR as evaluation metrics. Including additional metrics such as SSIM or VMAF would provide a more comprehensive assessment of video quality and better capture perceptual quality variations.
7. The analysis in Section 4.4 lacks depth, with insufficient explanation of Figures 6 and 11.
8. The layout of images in the paper is disorganized, making it difficult for readers to follow.

**Questions:**

1. Although the paper states that videos in the dataset were converted to a 224x224 resolution, it does not clarify the conversion method.
2. The paper references H.264 and H.265 as standards rather than specific codecs. It is unclear whether x264 or x265 was used for compression.

---

### Official Review · Reviewer_XNxB · 2024-11-11

**Soundness:** 3
**Presentation:** 3
**Contribution:** 3
**Rating:** 5
**Confidence:** 4

**Summary:**

This paper proposes MCUCoder, a lightweight image and video compression model specifically designed for IoT environments, utilizing an asymmetric computational architecture. MCUCoder achieves adaptive bitrate by sorting channels based on importance during the training stage and pruning less important channels during inference. It further leverages INT8 quantization to minimize power consumption and enhance processing speed. Experimental results demonstrate that MCUCoder significantly outperforms traditional M-JPEG in both compression efficiency and power consumption. Lack of Explicit Bitrate Allocation, and Limited RD Performance Comparison.

**Strengths:**

Strength: 1)Lightweight Design for IoT Devices. The encoder of MCUCoder is ultra-lightweight, with only 10.5k parameters and a memory footprint of 350kB, making it highly suitble for resource-constrained IoT devices. 2)Variable Bitrate. MCUCoder supports variable bitrate by generating a latent representation sorted by importance, allowing it to adapt efficiently to bandwidth-constrained environments. 3)Energy Efficiency. MCUCoder employs INT8 quantization, enabling it to apply on DSP accelerators and achieve energy efficiency comparable to M-JPEG.

**Weaknesses:**

Weakness: 1)Lack of Explicit Bitrate Allocation. The concept of pruning output channels to achieve variable bitrate has been studied extensively over the years [1, 2]. As the authors noted, different channels represent features at varying frequencies. However, feature frequency distribution can vary across an image, and simply discarding certain frequencies may significantly degrade specific regions, impacting both human and machine perception. In contrast, advanced video codecs typically employ sophisticated techniques that dynamically allocate bits based on the complexity of the image content. This level of granularity in bitrate allocation is something MCUCoder lacks. [1] Yang, F, Luis H, Yongmei C, and Mikhail G. M. “Slimmable Compressive Autoencoders for Practical Neural Image Compression.” CVPR 2021. [2] Tao, L., Gao, W., Li, G., Zhang, C. “Adanic: Towards practical neural image compression via dynamic transform routing.” ICCV 2023. 2)Limited RD Performance Comparison. The paper mainly compares MCUCoder to traditional methods like M-JPEG, H.264, and H.265, but lacks a thorough comparison with recent image/video compression methods specifically tailored for IoT or resource-constrained environments. This limits the paper's ability to demonstrate how MCUCoder stands against the latest advancements in image/video compression.

**Questions:**

Weakness: 1)Lack of Explicit Bitrate Allocation. The concept of pruning output channels to achieve variable bitrate has been studied extensively over the years [1, 2]. As the authors noted, different channels represent features at varying frequencies. However, feature frequency distribution can vary across an image, and simply discarding certain frequencies may significantly degrade specific regions, impacting both human and machine perception. In contrast, advanced video codecs typically employ sophisticated techniques that dynamically allocate bits based on the complexity of the image content. This level of granularity in bitrate allocation is something MCUCoder lacks. [1] Yang, F, Luis H, Yongmei C, and Mikhail G. M. “Slimmable Compressive Autoencoders for Practical Neural Image Compression.” CVPR 2021. [2] Tao, L., Gao, W., Li, G., Zhang, C. “Adanic: Towards practical neural image compression via dynamic transform routing.” ICCV 2023. 2)Limited RD Performance Comparison. The paper mainly compares MCUCoder to traditional methods like M-JPEG, H.264, and H.265, but lacks a thorough comparison with recent image/video compression methods specifically tailored for IoT or resource-constrained environments. This limits the paper's ability to demonstrate how MCUCoder stands against the latest advancements in image/video compression.

---

### Author Response · Authors · 2024-11-16
**Clarification on Baselines**

Thank you to all the reviewers for your constructive feedback.

We observed that the main concern across all reviews was the lack of IoT-specific baseline comparisons. We would like to clarify that MCUCoder is designed specifically for microcontrollers (MCUs) with extremely limited resources—typically around 1 to 2 MB of RAM. To the best of our knowledge, there are no other IoT-based encoder models we could use for comparison, except for M-JPEG. Reviewer YGWh suggested comparing with Distributed DVC [1]; however, this model operates at around 500 GFLOPs, making it impractical for MCUs. In contrast, MCUCoder only requires 0.003 GFLOPs, making it highly suitable for resource-constrained environments.

We would greatly appreciate it if the reviewers could suggest any specific models compatible with such hardware constraints that we might have missed.

[1] Low-complexity Deep Video Compression with A Distributed Coding Architecture.

---

### Note · Authors · 2024-11-23

I have read and agree with the venue's withdrawal policy on behalf of myself and my co-authors.